# Pgx: Hardware-Accelerated Parallel Game Simulators for Reinforcement Learning

**Sotetsu Koyamada**[1,2*]   **Shinri Okano**[1]   **Soichiro Nishimori**[3]
**Yu Murata**[1]   **Keigo Habara**[1]   **Haruka Kita**[1]   **Shin Ishii**[1,2]

[1]Kyoto University   [2] ATR   [3]The University of Tokyo

## Abstract

We propose Pgx, a suite of board game reinforcement learning (RL) environments written in JAX and optimized for GPU/TPU accelerators. By leveraging JAX's auto-vectorization and parallelization over accelerators, Pgx can efficiently scale to thousands of simultaneous simulations over accelerators. In our experiments on a DGX-A100 workstation, we discovered that Pgx can simulate RL environments 10-100x faster than existing implementations available in Python. Pgx includes RL environments commonly used as benchmarks in RL research, such as backgammon, chess, shogi, and Go. Additionally, Pgx offers miniature game sets and baseline models to facilitate rapid research cycles. We demonstrate the efficient training of the Gumbel AlphaZero algorithm with Pgx environments. Overall, Pgx provides high-performance environment simulators for researchers to accelerate their RL experiments. Pgx is available at https://github.com/sotetsuk/pgx.

**Backgammon**   **Chess**   **Go**   **Shogi**

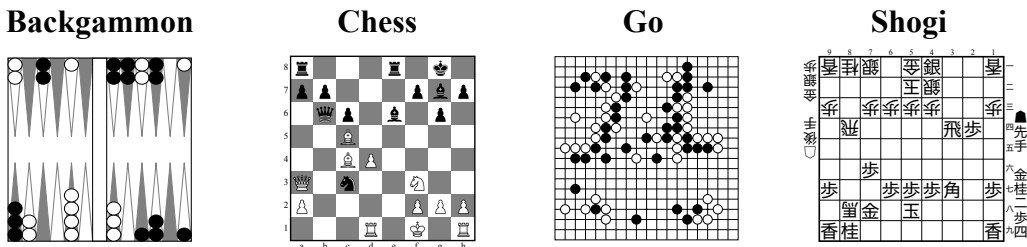

Figure 1: Example games included in Pgx.

## 1   Introduction

Developing algorithms for solving challenging games is a standard artificial intelligence (AI) research benchmark. Especially building AI, which can defeat skilled professional players in complex games like chess, shogi, and Go, has been a crucial milestone. Though reinforcement learning (RL) algorithms, which combine deep learning and tree search, are successful in obtaining such high-level strategies [1, 2, 3], complex games like chess and Go are still in the interest of AI research for developing RL and search algorithms in discrete state environments.

On the other hand, studying algorithms for solving large state-space games such as Go requires a huge sample size. MuZero [4], employing a learned model, has been successful in the domain of

---

*Correspondence to: Sotetsu Koyamada <koyamada@atr.jp>

37th Conference on Neural Information Processing Systems (NeurIPS 2023) Track on Datasets and Benchmarks.

game AI and can significantly reduce the number of interactions with the real environment. However, this does not render research on approaches without a learned model like AlphaZero unnecessary. Notably, it is mentioned that AlphaZero's learning is faster than MuZero's in the case of chess [5].

Thus, in RL research, a fast simulator of the environment, which achieves high throughput is often required. Simulators that possess practical speed and performance are often implemented in C++. In the machine learning community, however, Python serves as a *lingua franca*. Therefore, libraries like OpenSpiel [6] wrap the core C++ implementation and provide a Python API. However, this approach presents several challenges. Efficient parallelization of the environment is crucial for generating a large number of samples, but efficient parallel environments utilizing C++ threading, such as EnvPool [7], may not always be accessible from Python. In fact, to our knowledge, there are no libraries publicly available in Python that allow the use of efficient parallel environments for important game AI benchmarks like chess and Go from the identical Python API. Also, the RL algorithm often runs on accelerators (such as GPUs and TPUs), whereas simulation runs on CPUs, which makes additional data transfer costs between CPUs and accelerators.

In *continuous* state space environments, Brax [8] and Isaac Gym [9] demonstrate that environments that work on accelerators can dramatically improve the simulation throughput and RL training speed, resolving the data transfer and parallelization problems. Brax, written in JAX [10], a Python library, provides *hardware-accelerated* environments that leverage JAX's auto-vectorization, parallelization over accelerators, and Just-In-Time (JIT) compilation optimized for individual hardware platforms.

In this study, we offer the hardware-accelerated environments of complex, *discrete* state domains like chess, shogi and Go. Specifically, we introduce Pgx, a suite of efficient game simulators developed in JAX. Thanks to JAX's auto-vectorization and parallelization across multiple accelerators, Pgx can achieve high throughput on GPUs/TPUs. As of writing this paper, to our best knowledge, there is no other comprehensive game environment library written in JAX. Highlighted features of Pgx include:

- **Fast simulation**: Pgx provides high-performance simulators written in JAX that run fast on GPU/TPUs, similar to Brax. We demonstrated that Pgx is 10-100x faster than existing Python libraries such as PettingZoo [11] and OpenSpiel [6] on a DGX-A100 workstation (see Fig. 3).

- **Diverse set of games**: Pgx offers over 20 games, ranging from perfect information games like chess to imperfect information games like bridge (see Table 1). Pgx also offers miniature versions of game environments (e.g., miniature chess) to facilitate research cycles.

- **Baseline models**: Evaluating agents in multi-agent games is relative, requiring baseline opponents for evaluation. Since it is not always easy to have appropriate baselines available, Pgx provides its own baseline models. In Sec. 5, we demonstrate the availability of them with AlphaZero training.

Pgx is open-sourced and freely available at `https://github.com/sotetsuk/pgx`[2].

## 2 Related work

**Games in AI research.** An early study that combined neural networks (NNs) with RL to build world-class agents in a complex board game was TD-Gammon [12]. After the breakthrough of deep learning [13], RL agents combined with NNs performed well in the video game domain [14] and large state fully-observable board games, including chess, shogi, and Go [1, 2, 3]. RL agents with NNs also performed well in large-scale, partially observable games like mahjong [15]. However, these RL agents in complex board games require a huge number of self-play samples.

**Games as RL environment.** Game AI studies often have to pay high engineering costs, and there are a variety of libraries behind the democratization of game AI research. Arcade learning environment (ALE) made using Atari 2600 games as RL environments possible [16]. Several RL environment libraries provide classic board game suits [6, 11, 17]. Pgx aims to implement (classic) board game environments with high throughput utilizing GPU/TPU acceleration.

**Hardware-accelerated RL environments.** While hardware acceleration is a more specific approach compared to methods that run on CPUs, such as EnvPool [7], it has a major advantage of its

---
[2]See App. A for the license.

```python
import jax
import pgx

env = pgx.make("go_19x19")
init_fn = jax.jit(jax.vmap(env.init))
step_fn = jax.jit(jax.vmap(env.step))

batch_size = 1024
# Pseudo-random number generator keys determine the first players
rng_keys = jax.random.split(jax.random.PRNGKey(9999), batch_size)

state = init_fn(rng_keys)  # Vectorized initial states
while not (state.terminated | state.truncated).all():
  action = model(state.current_player, state.observation, state.legal_action_mask)
  state = step_fn(state, action)  # state.rewards with shape (1024, 2)
```

Figure 2: Basic usage of Pgx. The *init* function generates the initial *state* object. The *state* object has an attribute *current player* that indicates the agent which acts next. In this case, since we are using a batch size of 1024 for a 2-player game, *current player* is a binary vector whose size is 1024. Note that *current player* is independent of the colors (i.e., first player or second player). Here, *current player* is determined randomly using a pseudo-random number generator. The *step* function takes the previous *state* and an *action* vector, whose size is 1024, as input and returns the next *state*. The *observation* of the *current player* can be accessed through *state*. In the case of Go, for example, each *observation* has a shape of $1024 \times 19 \times 19 \times 17$. The available actions at the current *state* can be obtained through the boolean vector *legal action mask*. Here, it has a shape of $1024 \times 362$. For more detailed API description and usage, refer to the Pgx documentation at https://sotetsuk.github.io/pgx/.

ability to leverage accelerators for parallel execution, enabling high-speed simulations. Also, NN training is often performed on GPU/TPU accelerators, and there is an advantage that there is no data transfer cost between CPU and GPU/TPU accelerators. There is a wide range of environments available through various open-source software. In particular, JAX-based environments have gained popularity due to their high scalability over accelerators. These include:

- **gymnax** [18], which re-implements popular RL environments, including classic control (e.g., Pendulum and mountain car), bsuite [19] and a selection of environments from MinAtar [20].

- **Brax** [8], which provides a variety of continuous control tasks such as Ant and Humanoid.

- **Jumanji** [21], which offers RL environments for combinatorial optimization problems in routing (e.g., Travelling Salesman Problem; TSP), packing (e.g., bin-packing), and logic (e.g., 2048).

Pgx complements these environments by offering a (classic) *board game* suite for (multi-agent) RL research. Other hardware-accelerated environments include Isaac Gym [9] for continuous control, CuLE [22] as a GPU-based Atari emulator, and WarpDrive [23] for multi-agent RL research.

**Algorithms and architectures that can leverage Pgx.** The Anakin architecture [24] is an RL architecture that enables efficient utilization of accelerators and fast learning under the constraint that both the algorithm and environment are written as pure JAX functions. The architecture is capable of scaling up to (potentially) thousands of TPU cores with a simple configuration change. Since all Pgx environments are implemented using pure JAX functions, the Anakin architecture is applicable to any Pgx environment. Gumbel AlphaZero [5] improves the performance of AlphaZero when the number of simulations is small by employing the Gumbel-Top-k trick for search. They provide JAX-based Gumbel AlphaZero implementation[3] [25], which allows batch planning on accelerators. In Sec. 5, we use this implementation to show the example of Pgx usage in AlphaZero training.

---

[3] https://github.com/google-deepmind/mctx

Table 1: Available games in Pgx (as of v1.4.0).

| Env Name | # Players | Obs. shape | # Actions | Tag | Ref. |
|---|---|---|---|---|---|
| 2048 | 1 | $4 \times 4 \times 31$ | 4 | perfect info. (w/ chance) | [27, 28] |
| Animal shogi | 2 | $4 \times 3 \times 194$ | 132 | perfect info. | [29] |
| Backgammon | 2 | 34 | 156 | perfect info. (w/ chance) | [12, 27] |
| Bridge bidding | 4 | 480 | 38 | imperfect info. | [30, 31] |
| Chess | 2 | $8 \times 8 \times 119$ | 4672 | perfect info. | [3] |
| Connect Four | 2 | $6 \times 7 \times 2$ | 7 | perfect info. | [32] |
| Gardner chess | 2 | $5 \times 5 \times 115$ | 1225 | perfect info. | [33] |
| Go 9x9 | 2 | $9 \times 9 \times 17$ | 82 | perfect info. | [1, 2] |
| Go 19x19 | 2 | $19 \times 19 \times 17$ | 362 | perfect info. | [1, 2] |
| Hex | 2 | $11 \times 11 \times 4$ | 122 | perfect info. | [34, 35] |
| Kuhn poker | 2 | 7 | 4 | imperfect info. | [36, 6] |
| Leduc hold'em | 2 | 34 | 3 | imperfect info. | [37, 6] |
| MinAtar Asterix | 1 | $10 \times 10 \times 4$ | 5 | Atari-like | [20, 38] |
| MinAtar Breakout | 1 | $10 \times 10 \times 4$ | 3 | Atari-like | [20, 38] |
| MinAtar Freeway | 1 | $10 \times 10 \times 7$ | 3 | Atari-like | [20, 38] |
| MinAtar Seaquest | 1 | $10 \times 10 \times 10$ | 6 | Atari-like | [20, 38] |
| MinAtar Space Invaders | 1 | $10 \times 10 \times 6$ | 4 | Atari-like | [20, 38] |
| Othello | 2 | $8 \times 8 \times 2$ | 65 | perfect info. | [35] |
| Shogi | 2 | $9 \times 9 \times 119$ | 2187 | perfect info. | [3] |
| Sparrow mahjong | 3 | $11 \times 15$ | 11 | imperfect info. | |
| Tic-tac-toe | 2 | $3 \times 3 \times 2$ | 9 | perfect info. | [39] |

## 3 Pgx overview

In this section, we will provide an overview of the basic usage of Pgx, along with the fundamental principles behind the design of the Pgx API. Additionally, we will explain the overview of the game environments currently offered by Pgx as of this publication.

### 3.1 Pgx API design

Pgx does take inspiration from existing APIs, but it provides its own custom API for game environments. Specifically, two existing Python RL libraries highly inspired Pgx API:

- **Brax** [8], a physics engine written in JAX, providing continuous RL tasks, and
- **PettingZoo** [11], a multi-agent RL environment library available through Gym-like API [26].

Fig. 2 describes an example usage of Pgx API. The main difference from the Brax API comes from that the environments targeted by Pgx are multi-agent environments. Therefore, in Pgx environments, the next agent to act is specified as the *current player*, as in the PettingZoo API. On the other hand, the significant difference from the PettingZoo API stems from the fact that Pgx is a vectorization-oriented library. Therefore, Pgx does not use an *agent iterator* like PettingZoo and does not allow the number of agents to change. Concrete code examples comparing the Pgx API with the Brax and PettingZoo APIs are provided in App. B.

While this study does not focus on the design of the API, the Pgx API is sufficiently generic. At present, all game environments implemented in Pgx can be converted to the PettingZoo API and called from the PettingZoo API through the Pgx API (see App. B). This fact demonstrates the practical generality of the Pgx API. However, there is a limitation of the Pgx API in that it cannot handle environments where the number of agents dynamically changes. This limitation arises from the fact that the Pgx API is specialized for efficient vectorized simulation.

### 3.2 Available games in Pgx

The Pgx framework offers a diverse range of games, as summarized in Table 1. While Pgx primarily emphasizes multi-agent board games, it also includes some single-agent environments and Atari-

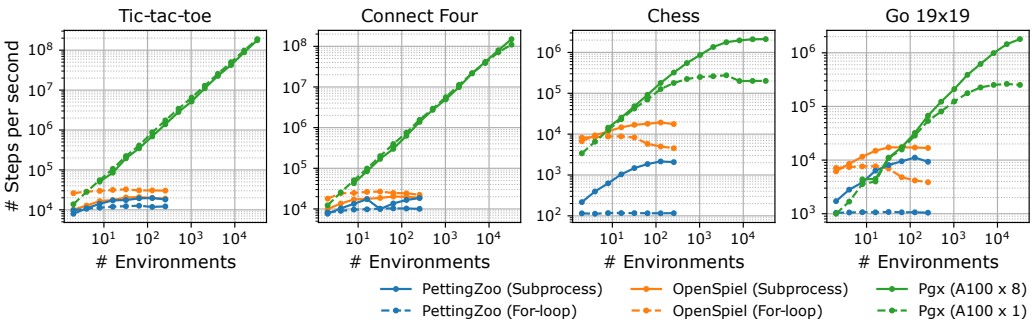

Figure 3: Simulation throughputs. Policies are random without learning processes. Error bars are not visible in this scale.

like environments to assist comprehensive RL research. Thus, as we will describe below, games implemented in Pgx span various categories, including two-player perfect information games, games with chance events, imperfect information games, and Atari-like games.

**Two-player perfect information games**  Pgx provides two-player perfect information games that range from simple games like *Tic-tac-toe* and *Connect Four* to complex strategic games like *chess*, *shogi*, and *Go 19x19*. While these traditional board games offer rich gameplay and strategic depth, they can be computationally demanding for many RL researchers. To address this, Pgx also includes smaller versions of shogi and chess: *Animal shogi* and *Gardner chess*. Although they have smaller board sizes compared to their original counterparts, these games are not mere toy environments. They retain enough complexity to provide engaging gameplay experiences for humans. Animal shogi, in particular, was specifically designed for children, while Gardner chess has a notable history of active play in Italy [33]. Additionally, Pgx offers other medium-sized two-player perfect information games such as *Go 9x9*, *Hex* and *Othello*.

**Games with (stochastic) chance events**  Pgx supports perfect information games with chance events, including *backgammon* and *2048*, which are popular benchmarks for RL algorithms in stochastic state transitions [27, 28]. These games introduce elements of randomness and uncertainty, adding a layer of complexity and decision-making under uncertainty to the gameplay.

**Imperfect information games**  In the realm of imperfect information games, Pgx provides several environments. These include *Kuhn poker*, *Leduc hold'em*, *Sparrow mahjong* (a miniature version of mahjong), and *bridge bidding*. These games involve hidden information, requiring agents to reason and strategize based on imperfect knowledge of the current game state.

**Atari-like games**  While the primary focus of Pgx is on board games, to make Pgx comprehensive and versatile, Pgx also implements *all* five environments from the MinAtar game suite: *Asterix*, *Breakout*, *Freeway*, *Seaquest*, and *Space Invaders* [20]. These Atari-like environments offer a more visually oriented and dynamic gameplay experience compared to traditional board games. Researchers often utilize MinAtar to conduct comprehensive ablation studies on RL methods in environments with visual inputs [38, 40]. Of these games, Freeway and Seaquest are highlighted as significant benchmarks for assessing the exploration capabilities of RL algorithms [38]. However, as of this writing, gymnax has not incorporated Seaquest into its suite.

For detailed descriptions of each environment, please refer to App. C.

## 4  Performance benchmarking: simulation throughput

Pgx excels in efficient and scalable simulation on accelerators thanks to JAX's auto-vectorization, parallelization over accelerators, and JIT-compilation. In this section, we validate it through experiments on an NVIDIA DGX-A100 workstation.

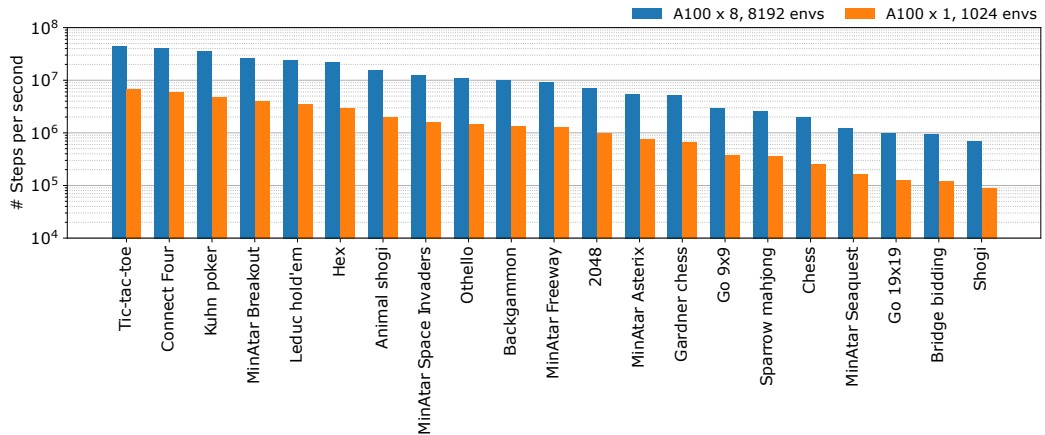

Figure 4: Simulation throughputs of all Pgx environments. Error bars are not visible in this scale.

## 4.1 Comparison to existing Python libraries

**Experiment setup.** We compare the pure simulation throughput of Pgx with existing popular RL libraries available in Python: PettingZoo [11] and OpenSpiel [6]. For the evaluation, we specifically selected Tic-tac-toe, Connect Four, chess, and Go, as these games are included in all three libraries. To our knowledge, neither PettingZoo nor OpenSpiel provides official parallelized environments from the Python API. Therefore, we prepared two implementations for each library:

1. **For-loop** (*DummyVecEnv*): sequentially executes and does not parallelize actually.
2. **Subprocess** (*SubprocVecEnv*): parallelize using the *multiprocessing* module in Python.

We modified and used *SubprocVecEnv* provided by Tianshou [41] for all competitor libraries. We performed all experiments on an NVIDIA DGX-A100 workstation with 256 cores; Pgx simulations used a single A100 GPU or eight A100 GPUs. We used random policies to evaluate the pure performance of simulators without agent learning. In all implementations, the environment automatically resets to the initial state upon reaching termination. The Pgx version used here was v0.8.0.

**Results.** Fig. 3 shows the results. We found that Pgx achieves at least 10x faster throughputs than other existing Python libraries when the number of vectorized environments (i.e., batch size) is large enough (e.g., 1024) on a single A100 GPU. Furthermore, when utilizing eight A100 GPUs, Pgx achieves throughputs of approximately 100x higher. This trend was identical from the simplest environment, Tic-tac-toe, to complex environments such as 19x19 Go.

## 4.2 Throughputs of other Pgx environments

The throughput of each environment is influenced by several factors, including the complexity and nature of the game, as well as the quality of its implementation. For instance, OpenSpiel demonstrates throughput of the same order for chess and Go 19x19. In contrast, PettingZoo's chess implementation exhibits a throughput approximately 10x slower than its Go 19x19 counterpart. This suggests that there might be room for optimization in PettingZoo's chess implementation regarding execution speed. To ensure that *all* Pgx environments achieve reasonable throughput and scalability like the four environments shown in Fig. 3, we measured the sample throughputs of all other Pgx environments on a DGX-A100 workstation, following the same approach as in the previous section. The number of vectorized environments was 1024 for a single A100 GPU and 8192 for eight A100 GPUs. The results are shown in Fig. 4. From these results, we can see that even in the slowest environment, Pgx achieves a throughput of approximately $10^5$ samples/second with a single A100 GPU. This demonstrates the efficiency of Pgx, considering that achieving such throughput with other Python libraries, as shown in the previous section, is challenging even in simpler environments like Tic-tac-toe. Furthermore, we observe a significant improvement in throughput when using eight A100 GPUs compared to a single A100 GPU in all environments. The throughput increased by an average of 7.4x across all

environments (at least 6.6x in the MinAtar Breakout environment). This highlights the excellent parallelization performance of Pgx across multiple accelerators.

# 5 AlphaZero on Pgx environments

In this section, we showcase the effectiveness of RL training on accelerators using the Pgx environments, with a specific focus on two-player perfect information games such as Go. For demonstrations of RL training in other game types, please refer to App. D. We begin by providing a brief overview of the Gumbel AlphaZero. Next, we describe the Pgx environments where we apply the Gumbel AlphaZero. Subsequently, we explain the experimental setup and discuss the selection of baseline models, which serve as anchor opponents for evaluation purposes. Finally, we present the results.

**AlphaZero [3] and Gumbel AlphaZero [5].** AlphaZero is an RL algorithm that leverages a combination of NNs and Monte Carlo Tree Search (MCTS). It has achieved state-of-the-art performance in chess, shogi, and Go, employing a unified approach. Through self-play, AlphaZero integrates MCTS with the current parameters of NNs, continually updating them through training on the samples generated during self-play. Gumbel AlphaZero is an adaptation of AlphaZero that removes several heuristics present in AlphaZero, enabling it to function even with a reduced number of simulations. In particular, Gumbel AlphaZero addressed the issue in the original AlphaZero where utilizing Dirichlet noise at the root node during tree search did not guarantee policy improvement. To overcome this limitation, Gumbel AlphaZero introduced an enhancement by utilizing the Gumbel-Top-k trick to perform sampling actions without replacement. They also proposed a MuZero version of the algorithm but we focus on the evaluation of AlphaZero in this paper. They released the Mctx library, which includes a JAX implementation of Gumbel AlphaZero. We utilized this library in our experiments. From now on, unless otherwise specified, when referring to "AlphaZero," it refers to Gumbel AlphaZero, not the original AlphaZero.

**Environments.** While Pgx enables fast simulations on accelerators, large-scale environments such as chess, shogi, and 19x19 Go pose significant challenges for researchers in terms of computational resources for learning. To address this, Pgx provides several small-scale environments, including miniature versions of shogi (Animal shogi) and chess (Gardner chess), as well as Hex (11x11) and Othello (8x8). In addition to these environments, we included the 9x9 Go environment to create a set of five environments for training AlphaZero. Here, we describe these environments briefly:

- **Animal shogi** is a 4x3 miniature version of shogi designed originally for children. Like shogi, players can reuse captured pieces. The small board size allows researchers to conduct research in an environment where planning ability is important with minimal computational resources.

- **Gardner chess** is a 5x5 variant of minichess that uses the leftmost five columns of the standard chessboard. It has a history of active play by human players in Italy [33].

- **Go 9x9** maintains the essential aspects of full-sized Go while being the smallest playable board size in the game. The advantage of 9x9 Go is that several full-sized Go AI models are also capable of playing on the 9x9 board, allowing us to use 9x9 Go as a reliable benchmark (e.g., [5]).

- **Hex** is a game played on an 11x11 board where two players take turns placing stones, and the player who forms a connected path from one side of the board to the other with their stones wins. Its rules are simple, making it relatively easy to interpret for researchers.

- **Othello**, also known as Reversi, is played on an 8x8 board. Players take turns placing stones and flip the opponent's discs that are sandwiched. The game concludes when neither player can make a valid move, and the player with the most discs on the board wins.

For more detailed information about each environment, please refer to App. C.

**Training setup.** We trained the models using the same network architecture and hyperparameters across all five environments. The network architecture is 6 ResNet blocks with policy head and value head, following the structure outlined in the original AlphaZero study [3] basically but with a smaller model size. During the self-play, 32 simulations were performed at each position for policy improvement. In each iteration, we generated data for 256 steps with a self-play batch size of 1024 (i.e., the number of vectorized environments). We then divided this data into mini-batches of

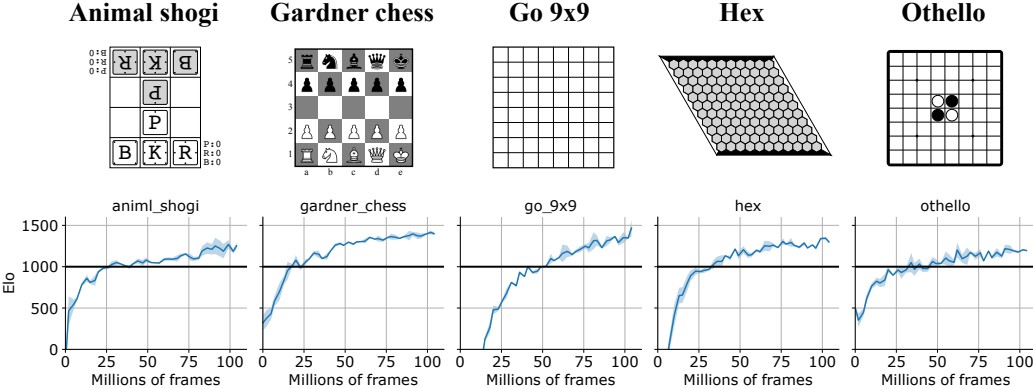

Figure 5: AlphaZero training results. Black line represents the Elo rating of baseline models provided by Pgx (1000 Elo). The shaded area represents the standard errors of two runs.

size 4096 for gradient estimation and parameter updates. We performed training for 400 iterations ($\approx$ 105M frames) in each environment. The choice of accelerator varied across the environments, but for example, in the case of 9x9 Go, we trained the model using a single A100 GPU, and the training process took approximately 8 hours. For more detailed information about the network architecture, hyperparameters, and accelerator specifications used in the training process, refer to App. E.

**Evaluation and baseline model selection.** We trained agents using the AlphaZero algorithm on the five environments described above and performed evaluations. However, in multi-agent games, the performance of the trained agents is relative, so we need a reference agent for comparison. However, finding a suitable baseline model for any environment is difficult, or even if it exists, it may not be computationally efficient. Therefore, for researchers and practitioners using Pgx, we created our own baseline models. It is important to note that our baseline models are not designed to be state-of-the-art or oracle models, but rather serve the purpose of examining the learning process within the Pgx environment. In the given learning setup, we selected the 200-iteration ($\approx$ 52M frames) model for 9x9 Go and the 100-iteration ($\approx$ 26M frames) model for other environments, considering their lower complexity compared to 9x9 Go. To evaluate the agents, we estimated the Elo rating through their pairwise matches. We used the *BayesElo* program to calculate the Elo rating[4][42]. We adjusted the Elo rating to ensure that our baseline models had 1000 Elo. During the evaluation matches, the agents conducted 32 simulations for each move like during the training.

**Results.** Fig. 5 presents the learning results of AlphaZero in the five environments. We can observe that the agents successfully learn in all five environments starting from a random policy with the same network architecture and hyperparameters. Furthermore, for the baseline model in 9x9 Go, we evaluated its performance by playing against Pachi [43] with 10K simulations per move, which was used as a baseline opponent in prior study [5]. The baseline model conducted 800 simulations for each move. Our baseline model outperformed Pachi with a record of 62 wins and 38 losses out of 100 matches, confirming its reasonable strength as a baseline model. Although no comparisons were made with other AIs in environments other than 9x9 Go, we trained them using the same network architecture and hyperparameters as in 9x9 Go. Given that the baseline model obtained in 9x9 Go exhibited reasonable strength throughout the learning process, we suppose that the baseline models in other environments, which were trained with exactly the same settings, have also learned reasonably. Therefore, we believe that researchers can accelerate their research cycles using the five environments and baseline models presented here, instead of relying on full-scale chess, shogi, and 19x19 Go, while exploring RL algorithms such as AlphaZero.

---

[4] https://www.remi-coulom.fr/Bayesian-Elo

# 6 Training scalability to multiple accelerators

In Sec. 4, we demonstrated a significant improvement in the pure throughput of Pgx when increasing the number of accelerators using a random agent. Here, we will show that increasing the number of cores also improves the learning speed in the AlphaZero training on Pgx environments.

**Experiment setup.** To demonstrate the improvement in learning speed by increasing the number of cores in AlphaZero training with Pgx, we conducted experiments in the 9x9 Go environment. In the experiments of Sec. 5, we performed training using a single A100 GPU in the 9x9 Go environment. Here, we conducted the exact same number of training frames but with an increased number of GPUs and batch size during self-play. In the experiment using a single A100 GPU, the batch size during self-play was set to 1024. However, in training with eight A100 GPUs, we increased the *self-play* batch size to 8192, which is eight times larger. It is important to note that the *training* batch size, learning rate, and the other hyperparameters were kept the same as in the single GPU case, ensuring that these hyperparameters did not affect the learning speed. Similar to Sec. 5, we used the baseline model as an anchor and adjusted Elo ratings so that the rating of the baseline model is 1000. Furthermore, in this setup, we want to mention that the time spent on self-play was dominant (more than 90%) compared to the time spent on training (gradient calculation and parameter updates).

**Results.** Fig. 6 shows the learning curves for the 9x9 Go environment using one A100 GPU and eight A100 GPUs. The shaded regions represent the standard error of runs with two different seeds. Based on the figure, it is evident that when both models are trained with the same number of training frames, the model trained with eight A100 GPUs achieves the same level of performance approximately four times faster than the model trained with a single GPU. This experimental outcome highlights

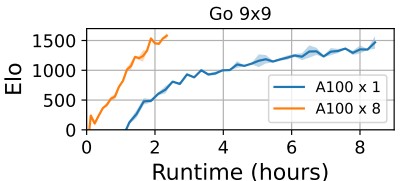

Figure 6: Multi-GPU training.

the fact that Pgx not only enhances throughputs in random play but also accelerates the learning process when training RL algorithms such as AlphaZero. These results underscore the practical utility of Pgx in the field of RL, providing researchers and practitioners with a powerful tool for efficient experimentation utilizing multiple accelerators.

# 7 Limitations and future work

There are several notable limitations users should take care of regarding Pgx, including:

- **Lack of support for Atari:** One of the limitations of Pgx is that it does not support Atari, which is an important benchmark in RL research. This limitation arises from the difficulty of implementing the Atari emulator and re-implementing the dynamics of each game in JAX.

- **Pgx API limitation:** While the games currently implemented in Pgx can be exported to the PettingZoo API, which is regarded as a general API for multi-agent games, Pgx API is not well-suited for handling certain types of games. These game types include those with a varying number of agents and those that involve chance players (nature players) such as poker.

- **JAX lock-in:** Although Pgx provides a convenient way to implement fast algorithms in Python without directly working with C++, it has a reliance on JAX, which may require users to be familiar with JAX. This can make it less straightforward to utilize other frameworks like PyTorch [44].

Our future work for Pgx includes the following:

- **Expansion of baseline algorithms and models:** Currently, we are unable to provide learning examples or models for large-scale games like chess, shogi, and Go 19x19, making it an important area for future work. We plan to expand the availability of strong models through proprietary training and connect with other strong AI systems to enhance the baselines.

- **Diversification of game types:** The current game collection in Pgx is biased towards (two-player) perfect information games. We plan to implement games with imperfect information, such as Texas hold'em and mahjong, to broaden the range of supported game types.

- **Verification on TPUs:** While we validated Pgx performance on an NVIDIA DGX-A100 workstation, it is important to conduct verification using Google TPUs as well. This will provide valuable insights into the performance and scalability of Pgx on different hardware architectures.
- **Human-vs-agent UI:** Developing a user interface that enables human-versus-agent gameplay is important future work. This will allow researchers with domain knowledge to conduct high-quality evaluations and experiments, fostering improved research and assessment.

By addressing these areas in our future work, we aim to enhance the capabilities and applicability of Pgx in the field of RL research and game AI for both researchers and practitioners.

## 8 Conclusion

We proposed Pgx, a library of hardware-accelerated game simulators that operate efficiently on accelerators, implemented in JAX. Pgx achieves 10-100x higher throughput compared to other libraries available in Python and demonstrates its ability to scale and train using multiple accelerators in the context of AlphaZero training. By providing smaller game environments, along with their baselines, Pgx facilitates the development and research of RL algorithms and planning algorithms that can operate at faster speeds. We anticipate that Pgx will contribute to advancing the field in terms of developing efficient RL algorithms and planning algorithms in these accelerated environments.

## Acknowledgments

This paper is based on results obtained from a project, JPNP20006, subsidized by the New Energy and Industrial Technology Development Organization (NEDO).

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

# A License

The source code of Pgx is available at https://github.com/sotetsuk/pgx under the Apache-2.0 license. However, since the original source code of the MinAtar game suite is released under the GPL 3.0 license, a separate extension for Pgx called *pgx-minatar* is provided at https://github.com/sotetsuk/pgx-minatar, which adheres to the GPL 3.0 license.

# B Comparison to Brax and PettingZoo APIs

The Pgx API draws inspiration from the Brax and PettingZoo APIs, and we illustrate this with actual code examples. Fig. 7 and Fig. 8 present the code examples for the Pgx and Brax APIs, respectively. While both APIs share similarities, the Pgx API handles multi-agent environments and the *current player* vector specifies which agents are to act. Similar to Pgx, the PettingZoo API (Fig. 9) designates the next agent to act using the *agent iterator* concept. However, since Pgx is a library centered on environment vectorization, we prefer a vectorized *current player* to a flexible but dynamic iterator.

```
1   import jax
2   import pgx
3
4   env = pgx.make("go_19x19")
5   init_fn = jax.jit(jax.vmap(env.init))
6   step_fn = jax.jit(jax.vmap(env.step))
7
8   batch_size = 1024
9   rng_keys = jax.random.split(jax.random.PRNGKey(9999), batch_size)
10
11  state = init_fn(rng_keys)
12  while not (state.terminated | state.truncated).all():
13      action = model(state.current_player, state.observation, state.legal_action_mask)
14      state = step_fn(state, action)
```

Figure 7: Example usage of Pgx API.

```
1   import jax
2   from brax import envs
3
4   env = envs.get_environment("ant")
5   reset_fn = jax.jit(jax.vmap(env.reset))
6   step_fn = jax.jit(jax.vmap(env.step))
7
8   batch_size = 1024
9   rng_keys = jax.random.split(jax.random.PRNGKey(9999), batch_size)
10
11  state = reset_fn(rng_keys)
12  while not state.done.all():
13      action = model(state)
14      state = step_fn(state, action)
```

Figure 8: Example usage of Brax API.

```
1   from pettingzoo.classic import go_v5
2
3   env = go_v5.env()
4   env.reset()
5   for agent in env.agent_iter():
6       observation, reward, terminated, truncated, info = env.last()
7       action = model(agent, observation["observation"], observation["action_mask"])
8       env.step(action)
9   env.close()
```

Figure 9: Example usage of PettingZoo API.

To prove the practical generality of the Pgx API, we provide a demonstration illustrating the conversion from Pgx API to PettingZoo API, available at https://github.com/sotetsuk/pgx/blob/main/colab/pgx2pettingzoo.ipynb.

# C   Game explanations

This section describes the short summary of each game implemented in Pgx (as of v1.4.0), except the MinAtar suite. See [20] for the description of the MinAtar suite. For the full description of each game, please refer to the Pgx documentation from https://github.com/sotetsuk/pgx.

- 2048
- Animal shogi
- Backgammon
- Bridge bidding
- Chess
- Connect Four
- Gardner chess
- Go

- Hex
- Kuhn poker
- Leduc hold'em
- Othello
- Shogi
- Sparrow mahjong
- Tic-tac-toe

## 2048

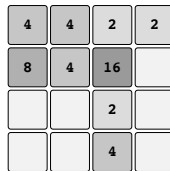

| Game | 2048 |
|---|---|
| Number of players | 1 |
| Observation shape | $4 \times 4 \times 31$ |
| Number of actions | 4 |
| Tag | perfect information with chance |

The game of 2048 [45] is a single-player perfect information game with chance events on a 4x4 board. The rules are simple, but to play well, agents need planning ability in stochastic dynamics.

**Rules.**   The objective is to create larger-numbered tiles by merging tiles. The player can take four actions: left, up, right, or down. All tiles move in the chosen direction, and when tiles with the same number collide, they merge, forming a new tile with twice the value (e.g., $4 + 4 = 8$). When a new tile is created, the new tile number is rewarded. After the tiles have moved, a new tile whose value is either two or four is randomly placed in empty positions. The probability of a two appearing is 0.9, and the probability of a four appearing is 0.1. The game ends when no legal move exists.

**Observation.**   The observation design follows [27]. Each plane represents a $4 \times 4$ board, and each tile number is encoded in a 31-bit binary representation.

**Action.**   Left (0), up (1), right (2), or down (3).

**Reward.**   The sum of merged tiles.

# Animal shogi

| Game | Animal shogi |
|------|-------------|
| Number of players | 2 |
| Observation shape | $4 \times 3 \times 194$ |
| Number of actions | 132 |
| Tag | perfect information |

Animal shogi is a miniature version of the traditional Japanese board game, shogi, played on a small 4x3 board. It is a two-player perfect information game. Originally developed for children, it possesses sufficient complexity for human play, making it more than just a toy environment.

**Rules.** Two players take turns moving their pieces, aiming to achieve *checkmate* – a situation where the king is attacked and cannot make a legal move to escape the threat. The pieces used in Animal shogi are *Lion* (**K**ing), *Giraffe* (**R**ook), *Elephant* (**B**ishop), and *Chick* (**P**awn). The available directions for movement are indicated by circular dots for each piece. Note that each piece can move only one square at a time, even if the piece is a Giraffe (Rook). The Chick (Pawn) can be *promoted* to a *Hen* (**G**old) by entering the opponent's territory (the farthest rank). Similar to ordinary shogi, captured opponent pieces can be reused by dropping them on the board. A player can win by checkmating the opponent's Lion (King) or by having their Lion (King) enter the opponent's territory (*Try rule*). If the same position occurs three times, it results in a repetition draw.

**Observation.** The observation design follows AlphaZero [3]. Each plane represents a $4 \times 3$ board and encodes the position-dependent features in Table 2. P1 represents the current player, and P2 represents the opponent player. As in AlphaZero, the last 8-step history is encoded ($24 \times 8 = 192$

Table 2: Animal shogi position-dependent features.

| Feature | # Planes |
|---------|----------|
| P1 piece | 5 |
| P2 piece | 5 |
| P1 prisoner piece count | 6 |
| P2 prisoner piece count | 6 |
| Repetitions | 2 |
| Total | 24 |

planes in total). Also, Table 3 shows the position-independent features. Each plane has the same values for all positions.

Table 3: Animal shogi position-independent features.

| Feature | # Planes |
|---------|----------|
| Color | 1 |
| Elapsed timesteps (normalized to $[0, 1]$) | 1 |

**Action.** Action consists of (1) the source position of the piece to move and the direction to move ($12 \times 8 = 96$), and (2) the position of the piece to drop and the type of piece to drop ($12 \times 3 = 36$), for a total of 132 discrete actions.

**Reward.** Each player gets +1 (win), −1 (lose), or 0 (draw).

# Backgammon

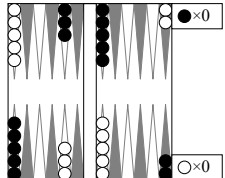

| Game | Backgammon |
|---|---|
| Number of players | 2 |
| Observation shape | 34 |
| Number of actions | 156 |
| Tag | perfect information with chance |

Backgammon is a two-player game with perfect information that also incorporates chance events. It serves as an important benchmark for RL in stochastic environments [12, 27]. To excel, agents require a high planning capability within stochastic environments.

**Rules.** Players, represented by white and black colors, aim to move their set of 15 *checkers* across a board consisting of 24 *points*. The objective is to be the first to *bear off* all their checkers, moving in the opposite direction. Each turn involves rolling two dice, determining the number of points a player can move their checkers. If both dice show the same number, the player can make four moves of that number. The game has several constraints on legal actions:

- Checkers cannot bear off until all of them have reached the *home board* (an area one-quarter of the distance from the goal).

- A point with two or more opponent's checkers stacked on it is *blocked*, and the player cannot move their checkers onto that point.

- By moving a checker to a point where the opponent has only one checker, the player can *hit* the opponent's checker and send it to the central bar. A player with checkers on the bar must first move those checkers before moving any other checkers.

The game ends when one of the players has borne off all their checkers. Victory rewards differ:

- A *gammon win* (2 points) is when the opponent has not borne off any checkers.

- A *backgammon win* (3 points) is a gammon win where the opponent has checkers left on the bar or within the winner's home board.

- All other victories are termed *single wins* (1 point).

**Observation.** Table 4 shows the backgammon features. The first 28 features are the same as in [27]. The last 6 features encode the number of playable die number.

Table 4: Backgammon features.

| Feature | Size |
|---|---|
| Checkers on points | 24 |
| Checkers on bar | 2 |
| Checkers borne off | 2 |
| Number of available moves for each die number | 6 |
| Total | 34 |

**Action.** Action in Pgx follows [27]. There are $26 \times 6 = 156$ discrete actions. Each action consists of 26 *source* positions and *die* number. The first source position is a *no-op* when there is no movable checker, the second source is the bar, and the remaining 24 sources represent each point on board.

**Reward.** Each player gets the game payoff as a reward.

# Bridge bidding

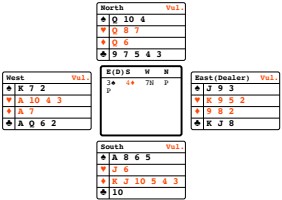

| Game | Bridge bidding |
|---|---|
| Number of players | 4 |
| Observation shape | 480 |
| Number of actions | 38 |
| Tag | imperfect information |

Contract bridge is an imperfect information game played by four people in teams of two. Cooperation within the team is required to win.

**Game flow.** In the contract bridge, after each player is dealt 13 cards, there are two phases: (1) *bidding* and (2) *playing*.

- **Bidding**: Each player bids in an auction format to determine the *contract*, the target number of *tricks* their team will try to achieve in the next playing phase.
- **Playing**: Each player plays a card, and the player who plays the "strongest" card wins the trick. This process is repeated 13 times, aiming to maximize the number of tricks won by the team and achieve the target number of tricks for their team or prevent the opponent team from achieving their target number of tricks.

Bidding is considered more challenging than playing and is believed to have a significant impact on the outcome of the game. Therefore, previous studies [46, 30, 31] have often focused only on the bidding part by replacing the playing part results using a double dummy solver[5]. Pgx also follows this setting.

**Rules of bidding.** The four players take turns to act. The available actions for each player are as follows: (1) *Bid* "higher" than the previous bid, (2) *Pass*, (3) *Double* the opponent's last bid, or (4) *Redouble* in response to the opponent team's double. There are 35 possible bid combinations, including the target trick number (1-7) and the suit (club, diamond, heart, spade, no trump). If three consecutive passes occur after someone's last bid, the bidding phase ends, and the bidding team, target number of tricks, and *trump* suit are determined (if no one bids and there are four consecutive passes, the game ends in a draw). The rewards are based on the achievement of the target, the magnitude of the target trick number, and whether a double or redouble is present.

**Observation.** Table 5 shows the bridge bidding features. Observation design follows [31, 6].

Table 5: Bridge bidding features.

| Feature | Size |
|---|---|
| Vulnerability | 4 |
| Pass before the opening bid | 4 |
| Bidding history (35 × 4 × 3) | 420 |
| Current player's hand | 52 |
| Total | 480 |

**Action.** There are 38 discrete actions: pass, double, redouble, and 35 bids.

**Reward.** Each player gets the game payoff as a reward.

---

[5] https://github.com/dds-bridge/dds

# Chess

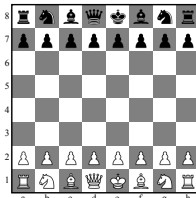

| Game | Chess |
|---|---|
| Number of players | 2 |
| Observation shape | $8 \times 8 \times 119$ |
| Number of actions | 4672 |
| Tag | perfect information |

Chess is a two-player perfect information game on an 8x8 board.

**Rules.** Two players take turns moving their pieces, aiming to achieve *checkmate* – a situation where the king is attacked and cannot make a legal move to escape the threat. Each piece has its own specific movement rules. While capturing the opponent's pieces is allowed, unlike in shogi, the captured pieces cannot be reused. In addition to checkmate, there are other terminal conditions for a draw, such as a threefold repetition of the same position or a *stalemate* (when the king is not in check but has no legal moves). There are also special moves called *pawn promotion*, *en passant*, and *castling*.

**Observation.** The observation design follows AlphaZero [3]. Each plane represents an $8 \times 8$ board and encodes the position-dependent features (Table 6). As in AlphaZero, the last 8-step history is

Table 6: Chess position-dependent features.

| Feature | # Planes |
|---|---|
| P1 piece | 6 |
| P2 piece | 6 |
| Repetitions | 2 |
| Total | 14 |

encoded ($14 \times 8 = 112$ planes in total). Also, Table 7 shows the 7 position-independent feature planes.

Table 7: Chess position-independent features.

| Feature | # Planes |
|---|---|
| Color | 1 |
| Total move count | 1 |
| P1 castling | 2 |
| P2 castling | 2 |
| No progress count | 1 |
| Total | 7 |

**Action.** Action design also follows AlphaZero [3]. There are $64 \times 73 = 4672$ discrete actions. Each action consists of 64 *source* positions and 73 moves, where 56 moves are queen moves, 8 moves are knight moves, and 9 moves are underpromotions.

**Reward.** Each player gets $+1$ (win), $-1$ (lose), or $0$ (draw).

## Connect Four

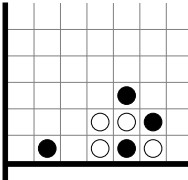

| Game | Connect four |
|---|---|
| Number of players | 2 |
| Observation shape | $6 \times 7 \times 2$ |
| Number of actions | 7 |
| Tag | perfect information |

Connect Four is a two-player perfect information game played on a 7x6 board.

**Rules.**  Players take turns dropping discs into any of the seven columns. The objective is to create a line of four of their own discs either vertically, horizontally, or diagonally. The player who achieves this first is declared the winner. If all the spaces on the board are filled, and neither player has managed to create a line of four discs, the game ends in a draw.

**Observation.**  Observation consists of two $6 \times 7$ feature planes (Table 8).

Table 8: Connect Four features.

| Feature | # Planes |
|---|---|
| P1 discs | 1 |
| P2 discs | 1 |

**Action.**  Each action represents the column index into which the player drops the token.

**Reward.**  Each player gets $+1$ (win), $-1$ (lose), or $0$ (draw).

## Gardner chess

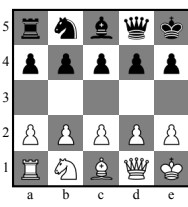

| Game | Gardner chess |
|---|---|
| Number of players | 2 |
| Observation shape | $5 \times 5 \times 115$ |
| Number of actions | 1225 |
| Tag | perfect information |

Gardner chess is a two-player perfect information game played on a 5x5 board. It uses the pieces corresponding to the leftmost 5 columns of standard chess.

**Rules.**  The rules are the same as in regular chess, except that *double pawn moves*, *en passant*, and *castling* are not allowed.

**Observation.**  The observation design follows AlphaZero [3]. Each plane represents a $5 \times 5$ board and encodes position-dependent features (Table 9). As in AlphaZero, the last 8-step history is

Table 9: Gardner chess position-dependent features.

| Feature | # Planes |
|---|---|
| P1 piece | 6 |
| P2 piece | 6 |
| Repetitions | 2 |
| Total | 14 |

encoded ($14 \times 8 = 112$ planes in total). Also, Table 10 shows position-independent feature planes.

Table 10: Gardner chess position-independent features.

| Feature | # Planes |
|---|---|
| Color | 1 |
| Total move count | 1 |
| No progress count | 1 |
| Total | 3 |

**Action.** Action design also follows AlphaZero [3]. There are $25 \times 49 = 1225$ discrete actions. Each action consists of 25 *source* positions and 49 moves, where 32 moves are queen moves, 8 moves are knight moves, and 9 moves are underpromotions.

**Reward.** Each player gets $+1$ (win), $-1$ (lose), or $0$ (draw).

## Go

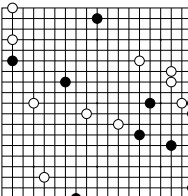

| Game | Go 9x9 | Go 19x19 |
|---|---|---|
| Number of players | 2 | |
| Observation shape | $9 \times 9 \times 17$ | $19 \times 19 \times 17$ |
| Number of actions | 82 | 362 |
| Tag | perfect information | |

Go is a two-player perfect information game played on a 19x19 board. The essential strategic aspects of Go can be preserved even on smaller boards like 9x9. There are variations of the Go rules, such as Chinese rules and Japanese rules. In computer Go, the Tromp-Taylor rules [47] are commonly used, and Pgx also follows them. To address the inherent advantage of the first player (black), it is common to add a scoring adjustment called *komi* (e.g., 6.5) to the final score of the second player (white). This improves fairness and helps to avoid a draw. Pgx uses a *komi* of 6.5 by default.

**Observation.** The observation design follows the AlphaGo Zero observation design [2]. Each plane represents a $9 \times 9$ (or $19 \times 19$) board and encodes the position-dependent features (Table 11). As

Table 11: Go features.

| # Planes | Description |
|---|---|
| P1 stones | 1 |
| P2 stones | 1 |

is the case with AlphaGo Zero, the last 8-step history is encoded ($2 \times 8 = 16$ planes in total). An additional plane encodes the color of the current player. This is necessary for the agent to know the *komi* information.

**Action.** Each action represents the position on the board to place a stone. The last action represents pass.

**Reward.** Each player gets $+1$ (win), $-1$ (lose), or $0$ (draw).

# Hex

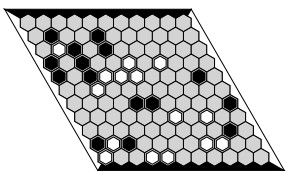

| Game | Hex |
|---|---|
| Number of players | 2 |
| Observation shape | $11 \times 11 \times 4$ |
| Number of actions | 122 |
| Tag | perfect information |

Hex is a two-player perfect information game played on an 11x11 board.

**Rules.** Players take turns placing stones on a board, aiming to connect one side of the board to the opposite side with their stones. A draw does not occur because both players cannot simultaneously connect their sides of the board. To balance the first-player advantage, the *swap rule* is implemented. This rule allows the second player, instead of placing a stone, to replace the color of the first player's stone with their own color at the mirrored position.

**Observation.** Each plane represents an $11 \times 11$ board and encodes position-dependent features (Table 12). The last two planes encode the color of the current player and whether the swap is a legal

Table 12: Hex position-dependent features.

| Feature | # Planes |
|---|---|
| P1 stones | 1 |
| P2 stones | 1 |

action (Table 13). Color information is necessary for the agent to know the side to connect.

Table 13: Hex position-independent features.

| Feature | # Planes |
|---|---|
| Color | 1 |
| Swap | 1 |

**Action.** The first 121 actions represent placing a stone on each cell of the board. The final action 121 is the swap action available only at the second turn.

**Reward.** Each player gets +1 (win) or −1 (lose).

# Kuhn poker

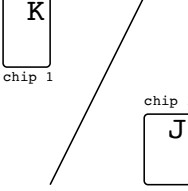

| Game | Kuhn poker |
|---|---|
| Number of players | 2 |
| Observation shape | 7 |
| Number of actions | 4 |
| Tag | imperfect information |

Kuhn poker is a two-player imperfect information game designed for research purposes [36].

**Rules.** The deck has three cards: Jack, Queen, and King. Each player is dealt one card, and the remaining card is unused. Players have four actions available: *check* (pass without placing a bet when no one else has bet), *call*, *bet*, and *fold*. The following scenarios can occur, with player A being the first to play and player B being the second:

- bet (A) - call (B): *Showdown*, and the winner takes +2.

- bet (A) - fold (B): A takes +1.
- check (A) - check (B): Showdown, and the winner takes +1.
- check (A) - bet (B) - call (A): Showdown, and the winner takes +2.
- check (A) - bet (B) - fold (A): B takes +1.

As Kuhn poker is a zero-sum game, the loser of the game receives the negative of the winner's payoff.

**Observation.** The observation consists of a binary vector of size 7 (Table 14).

Table 14: Kuhn poker features.

| Feature | Size |
| --- | --- |
| P1 hand | 3 |
| P1 chip | 2 |
| P2 chip | 2 |
| Total | 7 |

**Action.** Call (0), Bet (1), Fold (2) or Check (3).

**Reward.** Each player gets $+2$, $+1$, $-1$, or $-2$, depending on the game payoff.

## Leduc hold'em

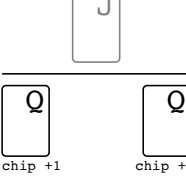

| Game | Leduc hold'em |
| --- | --- |
| Number of players | 2 |
| Observation shape | 34 |
| Number of actions | 3 |
| Tag | imperfect information |

Leduc hold'em is a two-player imperfect information game designed for research purposes [37].

**Rules.** The deck consists of six cards: two Jacks, two Queens, and two Kings. Each player starts with a bet of 1 chip. The game consists of two rounds. In the first round, each player is dealt one private card, and in the second round, one public card is revealed. In each round, players have the option to *call*, *raise*, or *fold*. If either player folds, the hand ends, and the opponent takes the *pot*. If both players call, the game proceeds to the next round. In the second round, it advances to *showdown*, where the winner is determined by the strength of their cards. Each player can raise 2 chips in the first round and 4 chips in the second round. In each round, each player is allowed to raise only once (a total of 2 raises per round). Therefore, the maximum number of chips that can be bet for each player is $1 + 2 \times 2 + 4 \times 2 = 13$.

**Observation.** The observation consists of a binary vector of size 34 (Table 15).

Table 15: Leduc hold'em features.

| Feature | Size |
| --- | --- |
| P1 hand | 3 |
| Public cards | 3 |
| P1 chip | 14 |
| P2 chip | 14 |
| Total | 34 |

**Action.** Call (0), Raise (1), or Fold (2).

**Reward.** Each player gets the game payoff as a reward.

## Othello

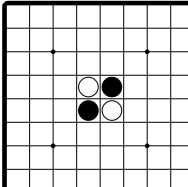

| Game | Othello |
|---|---|
| Number of players | 2 |
| Observation shape | $8 \times 8 \times 2$ |
| Number of actions | 65 |
| Tag | perfect information |

Othello is a two-player perfect information game played on an 8x8 board.

**Rules.** Players take turns placing discs. The player with more discs at the end wins. Players can place a disc in an empty position where it can sandwich the opponent's discs between their own discs, and the sandwiched discs are flipped to their own color. If a player has no valid move to make, they must pass. The game ends when neither player can make a legal move.

**Observation.** Each plane represents an $8 \times 8$ board and encodes the features in Table 16.

Table 16: Othello features.

| Feature | # Planes |
|---|---|
| P1 discs | 1 |
| P2 discs | 1 |

**Action.** The first 64 actions represent placing a disc on each square of the board. The last action represents pass.

**Reward.** Each player gets $+1$ (win), $-1$ (lose), or 0 (draw).

## Shogi

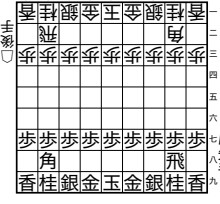

| Game | Shogi |
|---|---|
| Number of players | 2 |
| Observation shape | $9 \times 9 \times 119$ |
| Number of actions | 2187 |
| Tag | perfect information |

Shogi is a two-player perfect information game on a 9x9 board.

**Rules.** Two players take turns moving their pieces, aiming to achieve *checkmate* – a situation where the king is attacked and cannot make a legal move to escape the threat. Each piece has its own specific movement rules. Captured pieces can be reused by dropping them to the board (instead of moving a piece) on the player's turn. The game ends when a player achieves checkmate or when the game reaches a draw by four-fold repetition. Unlike chess, shogi does not have a draw by *stalemate* (when the king is not in check but has no legal moves). Not only pawns but also other pieces (except Gold and King) can *promote* by entering the opponent's territory (1-3 rows).

**Observation.** The observation design follows the *dlshogi* observation design [48]. Each plane represents a $9\times9$ board and encodes the position-dependent features in Table 17. Also, the observation

Table 17: Shogi position-dependent features.

| # Planes | Description |
|---|---|
| P1 piece | 14 |
| Attacked by P1 piece | 14 |
| Attacked by $N$ or more P1 pieces ($N = 1, 2, 3$) | 3 |
| P2 piece | 14 |
| Attacked by P2 piece | 14 |
| Attacked by $N$ or more P2 pieces ($N = 1, 2, 3$) | 3 |
| Total | 62 |

has the position-independent feature planes (Table 18).

Table 18: Shogi position-independent features.

| # Planes | Description |
|---|---|
| P1 hand | 28 |
| P2 hand | 28 |
| P1 king is checked | 1 |
| Total | 57 |

**Action.** The action design also follows the *dlshogi* action design [48]. There are $81 \times 27 = 2187$ distinct actions. Each action consists of 81 *destination* to which the piece moves and 27 *directions* from which the piece moves. The direction is one of 10 moves (8 King moves and 2 Knight moves), 10 moves with promotion, or 7 drops.

**Reward.** Each player gets $+1$ (win), $-1$ (lose), or $0$ (draw).

## Sparrow mahjong

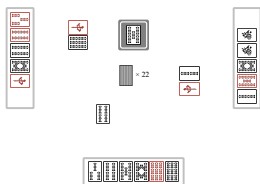

| Game | Sparrow mahjong |
|---|---|
| Number of players | 3 |
| Observation shape | $11 \times 15$ |
| Number of actions | 11 |
| Tag | imperfect information |

Sparrow mahjong is an imperfect information game played with 44 tiles. It is a simplified version of Japanese mahjong. Sparrow mahjong is designed for human players who are not familiar with the rules of full-size Japanese mahjong but requires the same essential skills as Japanese mahjong. It can be played with 2 to 6 players, but Pgx uses a 3-player version.

**Rules.** Sparrow mahjong has 11 types of tiles, and each tile has 4 copies, for a total of 44 tiles. The 11 types of tiles are *bamboo* 1-9 (1s-9s) and *red dragon* (rd), *green dragon* (gd). Each player starts with 5 tiles in their *hand*, and the game proceeds by each player drawing a tile from the deck and discarding one tile from their hand to a public *river*. The objective is to "accomplish" a hand of 6 tiles faster than other players and with a higher score. There are two ways to win the game: *ron* (winning by using a tile discarded by another player) and *tsumo* (winning by drawing a tile from the deck). A hand is "accomplished" when the hand consists of either (1) one *chow* (a sequence of three tiles) and one *pung* (a set of three identical tiles), (2) two *chows*, or (3) two *pungs*. For example, "2s 3s 4s 6s 6s 6s" (one *chow* and one *pung*), "1s 2s 3s 7s 8s 9s" (two *chows*), and "1s 1s 1s rd rd rd" (two

*pungs*) are accomplished hands. The accomplished hand is scored according to its difficulty. There is a minimum score required to win, and it is key to infer the opponent's hand. There is a unique rule in Japanese mahjong called *furiten*: a player cannot win by *ron* with a tile discarded previously by themselves. The game ends when a player wins or when the deck is empty.

**Observation.**    The observation consists of 15 feature planes, where each plane represents 11 tile types (Table 19). P1 represents the current player, while P2 and P3 represent the opponents.

Table 19: Sparrow mahjong features.

| Feature | # Planes |
|---|---|
| P1 hand | 4 |
| Red dora in P1 hand | 1 |
| Dora | 1 |
| All discarded tiles by P1 | 1 |
| All discarded tiles by P2 | 1 |
| All discarded tiles by P3 | 1 |
| Discarded tiles in the last 3 steps by P2 | 3 |
| Discarded tiles in the last 3 steps by P3 | 3 |
| Total | 15 |

**Action.**    Tile to discard.

**Reward.**    Each player gets the game payoff normalized to $[-1, 1]$.

## Tic-tac-toe

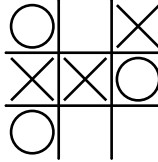

| Game | Tic-tac-toe |
|---|---|
| Number of players | 2 |
| Observation shape | $3 \times 3 \times 2$ |
| Number of actions | 9 |
| Tag | perfect information |

Tic-tac-toe is a two-player perfect information game played on a 3x3 board.

**Rules.**    Players take turns, with one marking X and the other marking O. The objective is for a player to place their mark in a vertical, horizontal, or diagonal line. The player who achieves this first is the winner. If all nine squares are filled and neither player has made a line, the game ends with a draw.

**Observation.**    Each plane represents a $3 \times 3$ board and encodes the features in Table 20.

Table 20: Tic-tac-toe features.

| Feature | # Planes |
|---|---|
| Marked by P1 | 1 |
| Marked by P2 | 1 |

**Action.**    Square index to place a mark.

**Reward.**    Each player gets +1 (win), −1 (lose), or 0 (draw).

# D    PPO training example

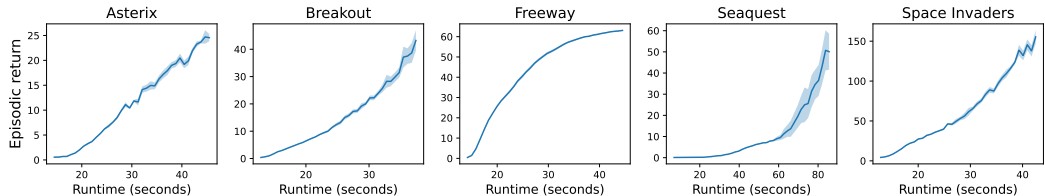

Figure 10: PPO training in MinAtar suite. Each model is trained up to 20M frames on a single A100 GPU. The shaded area represents the standard error of five runs.

In Sec. 5, we conducted learning experiments with AlphaZero for two-player, perfect information games such as 9x9 Go. Here, we present the results of RL training using the Proximal Policy Optimization (PPO) algorithm [49] in the MinAtar [20] environments as an example of model-free RL with Pgx.

MinAtar is a miniature version of Atari, specifically used by researchers to perform detailed ablation studies on RL methods using visual inputs (like Atari), without requiring large computational resources [38, 40]. MinAtar includes five games: Asterix, Breakout, Freeway, Seaquest, and Space Invader. It may be worth noting that Freeway and Seaquest are highlighted as possible useful benchmarks for exploration methods [38]. Pgx has re-implemented all five environments in JAX. The versions of *pgx* and *pgx-minatar* used here were v0.9.0 and v0.2.1, respectively.

**PPO implementation details.**    The implementation of PPO was based on the code available at https://github.com/luchris429/purejaxrl [50], with some modifications. As Pgx focuses on achieving high-speed training through vectorized simulation, we utilized a large batch size of 4096, similar to the training example in Brax [8]. We fine-tuned hyperparameters using the Asterix environment in the preliminary experiments (with a different seed used in the main experiments), and we maintained the same hyperparameters across all five games. The hyperparameters are listed in Table 21. Given the large batch size, the training was conducted up to 20M frames, which is longer compared to previous studies. However, it should be noted that the execution time is remarkably short, as discussed later.

Table 21: Hyperparameters in PPO training.

| Hyperparameter | Value |
|---|---|
| Rollout batch size (i.e., number of vectorized environments) | 4096 |
| Rollout length | 128 |
| Training minibatch size | 4096 |
| Number of epochs | 3 |
| Optimizer | Adam |
| Learning rate | 0.0003 |
| Gradient clipping max norm | 0.5 |
| Discount factor ($\gamma$) | 0.99 |
| GAE lambda ($\lambda$) | 0.95 |
| Clipping parameter ($\epsilon$) | 0.2 |
| Value function coefficient | 0.5 |
| Entropy coefficient | 0.01 |

**Results.**    Fig. 10 displays the learning results of PPO in the five MinAtar games. We conducted five learning runs with different seeds for each game. At each point of the learning process, we conducted 100 evaluation runs using the learned (stochastic) policy at that time and plotted the average score. The shaded area illustrates the standard error of the mean scores over the five runs. The runtime includes the RL training time and JIT-compilation time of JAX code. In all games, PPO with Pgx achieves sufficiently high scores in less than a minute, except for Seaquest, which takes more than 80 seconds to elapse 20M frames. These results demonstrate the effectiveness of vectorized execution in other game types than two-player, perfect information games implemented in Pgx.

# E  AlphaZero training details

This section describes the details of Gumbel AlphaZero training (Sec. 5). Our study fundamentally follows the original AlphaZero [3] framework except for the following points:

- To enable training on a single GPU, we alternated between self-play and learning, rather than executing them as two independent processes.
- To examine the impact of self-play speed on AlphaZero training speed, we discarded data after a single use, instead of employing a replay buffer.
- We utilized ResNet v2 [51] in place of ResNet v1.

Table 22 shows the hyperparameters used in the training. Note that we used the same hyperparameters for all five games. The GPUs used for training vary depending on the game. Table 23 shows the GPUs and runtime for each game. The version of Pgx used in the training is v0.8.0.

Table 22: Hyperparameters in AlphaZero training.

| Hyperparameter | Value |
|---|---|
| Number of residual blocks | 6 |
| Number of channels of conv. layer | 128 |
| Self-play batch size (i.e., number of vectorized environments) | 1024 |
| Self-play length in each iteration | 256 |
| Number of simulations per move | 32 |
| Training minibatch size | 4096 |
| Optimizer | Adam |
| Learning rate | 0.001 |
| Completed Q-values *value scale* | 0.1 |
| Completed Q-values *rescale values* | False |

Table 23: GPUs used in AlphaZero training.

| Environment | GPUs | Runtime (hours) |
|---|---|---|
| Animal shogi | A4000 x 1 | 6.2 |
| Gardner chess | A4000 x 4 | 14.3 |
| Go 9x9 | A100 x 1 | 8.6 |
| Hex | A4000 x 1 | 17.6 |
| Othello | A4000 x 1 | 11.4 |

