# OpenReview forum: "Pgx: Hardware-Accelerated Parallel Game Simulators for Reinforcement Learning"
_NeurIPS.cc/2023/Track/Datasets_and_Benchmarks — NeurIPS 2023 Datasets and Benchmarks Poster_

### Official Review · Reviewer_e7SA · 2023-07-20
**Highly useful GPU-based RL simulator**

**Rating:** 9
**Confidence:** 4
**Correctness:** Yes.
**Clarity:** Yes.

**Strengths:**

The paper is well-written and has the following strengths:
* **Make board game RL affordable**: previously, conducting research in board games with RL has been prohibitively expensive. For example, AlphaZero reported using thousands of TPUs for hours and days [1]. As a result, many researchers would encounter significant difficulties studying board game RL. PGX simulates board games several orders of magnitude faster than existing offerings and, therefore, can lower the barrier of entry for board game RL research.
* **Demonstrate successful end-to-end pipeline**: the author is demonstrated successful end-to-end training experiments across several board games, which offers solid open-source baselines for future research. With 8 A100s, the authors can train a highly effective Go 9x9 agent in just 2 hours, which is fast and affordable. Using Lambda Lab [2]'s pricing, 8 A100s costs $12 / hour and researchers can reproduce their work with just $24.


[1] https://arxiv.org/pdf/1712.01815.pdf

**Additional Feedback:**

Figure 6 seems somewhat confusing. Why is the line flat before 2 hours for the single A100 setting?

**Documentation:**

Yes.

**Limitations:**

Yes.

**Opportunities For Improvement:**

I am curious about the end-to-end training performance under Go 19x19 and how much it would cost to replicate alpha zero-level agent.

**Relation To Prior Work:**

Yes.

**Summary And Contributions:**

The authors present PGX, a GPU-based simulator with a focus on board games. PGX is written in JAX and can leverage auto-vectorization and Just-In-Time (JIT) to improve the throughput of RL environment simulation significantly. In particular, PGX can simulate games such as Go 19x19 and Chess several orders of magnitude faster than existing offerings in OpenSpiel (C++-based) and PettingZoo. In addition, the authors also use Gumbel AlphaZero to demonstrate successful end-to-end training experiments across several board games.

PGX offers solid RL baselines in board games trained with AlphaZero and can significantly lower the barrier of entry to do RL research in board games.

---

> ### Author Response · Authors · 2023-08-21
> **Author response to reviewer e7SA**
>
> Thank you for your insightful feedback on our paper.
>
> > I am curious about the end-to-end training performance under Go 19x19 and how much it would cost to replicate alpha zero-level agent.
>
> We recognize the significance of this query and are actively working on it. Once we have the results, we will promptly update our repository.
>
> > Figure 6 seems somewhat confusing. Why is the line flat before 2 hours for the single A100 setting?
>
> The appearance of a flat line is due to the truncation of the learning curve, which exceeds the y-axis limits. It's important to clarify that the curve isn't actually flat. The Elo rating system we used measures scores relative to a baseline Elo (1000 in our paper). Occasionally, the initial portion of the learning curve may be truncated due to this. In response to your feedback, we've expanded the y-axis range, though some truncation still remains.

---

> > ### Comment · Reviewer_e7SA · 2023-08-25
> > **Response to the authors**
> >
> > I thank the authors for the reply and look forward to seeing the results on Go 19x19 in the repo.

---

> > > ### Author Response · Authors · 2023-08-25
> > >
> > > Thank you again for the review. We will continuously develop and improve the experiments and baselines.

---

### Official Review · Reviewer_fFc4 · 2023-07-21
**Good paper but need more insights on performance/scalability on multiple GPUs**

**Rating:** 7
**Confidence:** 4

**Strengths:**

+ GPU accelerated of popular environments for RL and multi-agent research
+ Open source and provide baselines implementation for these games.
+ Easy to use API and interfaces to develop or integrate RL/MARL algorithms


**Additional Feedback:**

I would love authors add couple of real world environments to this benchmarks. Hardware accelerated environments are needed for these domain esp when applying RL or multi-agent RL.

**Clarity:**

Yes, it is reasonably easy to read. I did not have any difficulty in understanding problem or approach the authors have taken in this work.

**Correctness:**

Though mostly I feel the benchmarking is correct, I think from performance optimization wise there is more to do. Especially on revealing the bottlenecks associated from scaling from 1 GPU to 8 GPU. What is limiting this scalability is not answered in this paper.

**Documentation:**

Yes, there is a healthy documentation and active participation on GitHub.

**Limitations:**

Though this work is definitely welcome in educating researchers on the need for hardware accelerated environments especially for RL/multi-agent algorithms.  Though this work also focuses on games, real world domains where RL/multi-agent RL can play a huge role are incredibly slow. It would be nice if the authors take some example of real world and outline how they can use similar design methodology to accelerate those domains? I would love to see some general principles or methodology in constructing hardware accelerated environments in this paper.

**Opportunities For Improvement:**

+ While this is a much needed work on educating and providing hardware accelerated environments for researchers, it would be worth understanding the performance issues esp with the results.

 + For instance, in Fig 3, what is the reason why does the throughput (steps/sec) plateau for chess with the number of parallel environments whereas for others it does not?

+ In some cases, why is there a drop in throughput? (e.g., chess in pgx-A100 x 1). Also we see this trend for the baselines (either petting zoo or openSpiel as well?). Can you explain or throw some insights as to how these can be improved further?

+ Fig 4, how many parallel environments were used to collect this data?
Related to Fig 4, ideally one would expect some nice scaling from 1 GPU to 8 GPU. I understand it can't be always a linear scaling. In your case, the scaling seems to be poor. Do you have any data on how this looks as you scale from 1 GPU to 8 GPU? Also, can you tell us or the users of your codebase on how they can improve as they scale to something even larger?


**Relation To Prior Work:**

Yes, the authors have done a good job on citing and comparing prior works.

**Summary And Contributions:**

Pgx is a hardware-accelerated environment for a suite of board games, written in Jax. It offers several board game environments, including chess, go, and backgammon, along with relevant baselines. By leveraging auto-vectorization and JIT, Pax has been able to achieve a speedup of 10x-100x when run on a GPU.

---

> ### Author Response · Authors · 2023-08-21
> **Author response to reviewer fFc4**
>
> Thank you for your insightful review comments.
>
> > For instance, in Fig 3, what is the reason why does the throughput (steps/sec) plateau for chess with the number of parallel environments whereas for others it does not?
>
> The answer to this question largely depends on the game. For board games like chess and Go, the computation of the `legal action mask` often dominates. This is not unique to Pgx but is also the case in OpenSpiel and PettingZoo. The challenge arises because it is often difficult to determine if an action is legal without actually taking it. This requires making transitions for each possible action and then evaluating the resulting board state. While Pgx vectorizes these evaluations, this remains a bottleneck. This is expensive especially given the large size of the action space (4672 for chess, 362 for 19x19 Go). For instance, in chess, if we exclude the legal action computation, state transitions due to piece movements are straightforward and immediate. In contrast, games like Tic-tac-toe or Connect Four, where legal actions are obvious without actual transitions, scale well. However, calculating legal actions is almost essential in board games. Skipping this calculation might speed up environment transitions but could significantly slow down learning.
>
>
> > In some cases, why is there a drop in throughput? (e.g., chess in pgx-A100 x 1). Also we see this trend for the baselines (either petting zoo or openSpiel as well?). Can you explain or throw some insights as to how these can be improved further?
>
> Generally speaking, for a given computational resource, there exists an optimal batch size or optimal number of vectorized environment. Once you surpass this optimal point, throughput tends to degrade. In the case of Pgx, as previously mentioned, the bottleneck is the calculation of the legal action mask. While skipping this may allow us to shift the peak to a larger batch size, it does not mean we can infinitely increase the batch size.
>
> > Fig 4, how many parallel environments were used to collect this data? Related to Fig 4, ideally one would expect some nice scaling from 1 GPU to 8 GPU. I understand it can't be always a linear scaling. In your case, the scaling seems to be poor. Do you have any data on how this looks as you scale from 1 GPU to 8 GPU? Also, can you tell us or the users of your codebase on how they can improve as they scale to something even larger?
>
> We used 1024 and 8192 parallel environments for the data collection. As a response to this comment: (1) we have clarified the description related to this point in the manuscript, and (2) we've added these numbers to the figure's legend.
>
> We do not think the scale is poor. Please note that the y-axis is log-scale.
> It scales `7.42` times in average (`6.61` at worst in MinAtar breakout).
> Given your feedback, we added horizontal sub-grid lines in the figure and emphasizes its log-scale.
>
> > Though this work also focuses on games, real world domains where RL/multi-agent RL can play a huge role are incredibly slow. It would be nice if the authors take some example of real world and outline how they can use similar design methodology to accelerate those domains? I would love to see some general principles or methodology in constructing hardware accelerated environments in this paper.
>
> > I would love authors add couple of real world environments to this benchmarks. Hardware accelerated environments are needed for these domain esp when applying RL or multi-agent RL.
>
> Your comments are insightful and raise important points. While real-world environments might not always be accelerated using the same approach as Pgx, understanding when and how hardware acceleration is possible, and the associated methodologies, is crucial. However, these insights are often empirical and implementation-dependent, making it challenging to definitively include in the paper.
>
> > Though mostly I feel the benchmarking is correct, I think from performance optimization wise there is more to do. Especially on revealing the bottlenecks associated from scaling from 1 GPU to 8 GPU. What is limiting this scalability is not answered in this paper.
>
> Please refer to the response above.

---

> > ### Comment · Reviewer_fFc4 · 2023-08-22
> > **Thanks for clarification. Good improvments.**
> >
> > I would like to thank the authors for addressing and clarifying the issues related to data, performance, and throughput scaling in the paper. The updates made to the paper have also enhanced its readability and have effectively addressed the concerns raised above.
> >
> > Just to clarify any misunderstandings, I assume that the codebase and benchmark that will be utilized by the community and will continue to expand over time. In my previous comment, I merely suggested that the authors consider the inclusion of real-world tasks in the benchmark as a potential future addition, rather than a requirement for the current paper revision,

---

> > > ### Author Response · Authors · 2023-08-22
> > >
> > > Thank you for acknowledging the revisions and for clarifying your stance on the inclusion of real-world tasks. We appreciate your suggestion and will refine our codebase to assist RL researchers in addressing real-world challenges in the future.

---

### Official Review · Reviewer_5n9R · 2023-07-21
**Hardware Accelerated RL Environment for Board Games**

**Rating:** 7
**Confidence:** 3
**Correctness:** Seems correct.
**Clarity:** Yes.

**Strengths:**

+ This paper introduces an environment for RL that supports hardware acceleration out of the box
+ The library is well designed and accompanied by good documentation
+ The experiments conducted demonstrate the advantageous impact of hardware acceleration

**Additional Feedback:**

N/A

**Documentation:**

Yes.

**Ethics:**

No.

**Limitations:**

The limitations section clearly lays out the steps for improvement.

**Opportunities For Improvement:**

The limitations section clearly lays out the steps for improvement. Completion of some of those before submission would have made the submission much stronger.

**Relation To Prior Work:**

Yes and other recent hardware accelerated RL libraries are also mentioned and the differences are noted.

**Summary And Contributions:**

This paper presents a novel RL environment for board games, leveraging hardware acceleration. The implementation in JAX allows smooth deployment across CPUs, GPUs, and TPUs, facilitating extensive scalability. Moreover, the GitHub repository exhibits good design, accompanied by a highly informative README and companion website, streamlining the adoption process for potential users.

The reviewer is unable to give it the highest score for a few reasons. First and foremost, only supporting standard board games limits the usefulness of the environment. While, exciting additions are on the roadmap, they are not implemented yet. Compounding that is the limitation of the API which doesn't make it seem very extensible. That makes the impact of the environment likely limited. Second, figures could be improved to increase the clarity and obviousness of impact. For example, Figure 5 could plot the baseline Elo on the graphs and could train the same model using a different environment and show that it performs worse (or takes a much longer time). Not enough attention is given to Figure 3, which I think is your headline result. The purpose of Figure 4 is unclear, especially given the existence of Figure 3 and 6. It likely would be clearer as a table of speedups from 1 to 8 GPUs instead of the bars where the differences are hard to appreciate given how compressed the figure is.

---

> ### Author Response · Authors · 2023-08-21
> **Author response to reviewer 5n9R**
>
> Thank you for your constructive feedback.
>
> > First and foremost, only supporting standard board games limits the usefulness of the environment. While, exciting additions are on the roadmap, they are not implemented yet.
> > Compounding that is the limitation of the API which doesn't make it seem very extensible. That makes the impact of the environment likely limited
>
> As stated in our limitations section, Pgx is not a universal Swiss Army knife that can adapt to any environment. However, we believe that not every library needs to be all-encompassing.
> Pgx can complement libraries like Gymnax, Brax, and Jumanji, which implement domains outside of games, and vice versa.
> For environments with dynamic agent changes, we believe a different, more suitable API should be developed for that purpose. While our API and its applications have limitations, we believe these limitations themselves make Pgx specialized and user-friendly for the board game domain.
>
> > Second, figures could be improved to increase the clarity and obviousness of impact. For example, Figure 5 could plot the baseline Elo on the graphs and could train the same model using a different environment and show that it performs worse (or takes a much longer time). Not enough attention is given to Figure 3, which I think is your headline result. The purpose of Figure 4 is unclear, especially given the existence of Figure 3 and 6. It likely would be clearer as a table of speedups from 1 to 8 GPUs instead of the bars where the differences are hard to appreciate given how compressed the figure is.
>
> Based on your comments, we have made the following changes to the manuscript:
>
> - We have plotted the baseline Elo in Fig. 5.
> - To ensure adequate attention is given to Fig. 3, which we believe showcases our primary results, we have added references to this figure in the section where we enumerate our contributions in the introduction.
> - Regarding Fig. 4, we have provided additional explanations within the manuscript to clarify its intent. We also had concerns that the y-axis being in log-scale might have been overlooked, so we added horizontal sub-grids to the figure to emphasize its log-scale.
>
> Regarding the AlphaZero training across different frameworks, since the training code utilizes the JAX-native environment implementation, it's challenging to adapt it to other frameworks.

---

> > ### Comment · Reviewer_5n9R · 2023-08-21
> > **Good Improvements**
> >
> > Thank you for your feedback and improvements. This makes your contributions much clearer and defends you design decisions well. I think your updates in response to other reviewers comments also made the paper much stronger.

---

> > > ### Author Response · Authors · 2023-08-22
> > >
> > > Thank you for acknowledging the revisions. We greatly appreciate your constructive feedback, which was instrumental in enhancing the clarity and strength of the paper.

---

### Official Review · Reviewer_uMKG · 2023-07-21
**PGX: A JAX-based Hardware-Accelerated Environment Suite**

**Rating:** 6
**Confidence:** 5
**Clarity:** Yes

**Strengths:**

This work boasts a robust codebase and impressive throughput benchmarks, along with informative scaling plots that draw comparisons to other existing Python approaches under identical settings.

**Additional Feedback:**

No

**Correctness:**

In the abstract:

> We discovered that PGX can simulate RL environments 10-100x faster than existing Python RL libraries.

I would recommend revising this sentence to:

> We discovered that PGX can simulate RL environments 10-100x faster than existing Python implementations.

This adjustment clarifies the comparison by specifically referring to "Python implementations" rather than "Python RL libraries" (Tianshou in this case), which could be confusing due to its broader implications, encompassing both vectorized environment implementations and RL algorithms.

**Documentation:**

The environment documentation for certain games, including Sparrow Mahjong, Bridge Bidding, Shogi, and Gardner Chess, is found to be missing or incomplete.

It would be highly beneficial for the authors to consider including a step-by-step tutorial to effectively demonstrate and explain a single pgx environment, thus enhancing usability for the research community.

**Limitations:**

A minor aspect to consider: the time required for environment resets could significantly impact the throughput of vectorized environments. It would be valuable for the authors to address this point in the paper to provide a more comprehensive understanding of the potential performance implications.

**Opportunities For Improvement:**

It would be nice for the authors to include additional end-to-end RL experiment results in the experimental section. For example, utilizing the same JAX PPO implementation to run Go 19x19 with PGX and PettingZoo environments and plotting the reward curve against wall time could provide more comprehensive insights.

It is recommended that the authors consider translating the code comments from Japanese to English to facilitate broader understanding and accessibility for the wider research community.

**Relation To Prior Work:**

Yes

**Summary And Contributions:**

This work presents PGX, a hardware-accelerated reinforcement learning (RL) environment suite developed using JAX. The suite features a variety of environments, encompassing MinAtar and a range of chess games, and caters to both single-player and multi-player settings. PGX showcases a remarkable improvement in performance, achieving at least a 10x speedup compared to existing Python-based implementations. Moreover, it can linearly scale up to 10k environments, achieving substantial training throughput on a single DGX-A100 machine.

---

> ### Author Response · Authors · 2023-08-21
> **Author response to reviewer uMKG**
>
> Thank you for your constructive feedback.
>
> > It would be nice for the authors to include additional end-to-end RL experiment results in the experimental section. For example, utilizing the same JAX PPO implementation to run Go 19x19 with PGX and PettingZoo environments and plotting the reward curve against wall time could provide more comprehensive insights.
>
> Thank you for the valuable suggestion. We are in the process of conducting more extensive RL experiments and benchmarking. Upon completion, we will update the manuscript or our repository with the results accordingly.
>
> > It is recommended that the authors consider translating the code comments from Japanese to English to facilitate broader understanding and accessibility for the wider research community.
>
> We've translated the in-code comments to English (except some specific terms like shogi piece names).
>
> > A minor aspect to consider: the time required for environment resets could significantly impact the throughput of vectorized environments. It would be valuable for the authors to address this point in the paper to provide a more comprehensive understanding of the potential performance implications.
>
> In our experiments, we've used auto-resetting. This means that in a vectorized environment, it resets immediately upon termination. We've added a sentence in our paper to clarify this.
>
> > In the abstract:
> >  ...
> >  I would recommend revising this sentence to:
> >  ...
>
> Thank you for the detailed feedback on the abstract. We've adjusted the phrasing to "existing implementations available in Python" to more accurately reflect that OpenSpiel provides a Python API by wrapping the C++ implementation.
>
> > The environment documentation for certain games, including Sparrow Mahjong, Bridge Bidding, Shogi, and Gardner Chess, is found to be missing or incomplete.
>
> We've enhanced and completed the documentation for these games.
>
> > It would be highly beneficial for the authors to consider including a step-by-step tutorial to effectively demonstrate and explain a single pgx environment, thus enhancing usability for the research community.
>
> We recognize the value of a comprehensive tutorial. While we currently offer a hands-on experience with the Pgx API via Colab, expanding on this tutorial is on our roadmap, and we'll be addressing it in the near future.

---

### Official Review · Reviewer_bSTx · 2023-07-21
**PGx is a strong contribution, however, the paper exhibits shortcomings.**

**Rating:** 7
**Confidence:** 4

**Strengths:**

Simulated environments are of utmost importance to the RL community, and PGx is likely to be of importance to many researchers. I believe the PGx library is a substantial contribution to the Benchmarks and Dataset Track. It will be valuable for the research community, providing efficient simulators for a suite of 20 diverse, discrete games.

The compatibility with PettingZoo API is neat. The authors provide a decent comparison to PettingZoo and OpenSpiel, and an in-depth use-case of training AlphaZero on 5 smaller environments which is informative.

**Additional Feedback:**

---- Updated score 6 to 7 post revisions. ----

**Clarity:**

The paper is clear and easy to follow. I particularly like Table 1. However, the motivation for the library is not clear (as discussed above). There is also a lack of details on specific API details. A reader would have to look through the examples in the repository to understand more.

**Correctness:**

There are sufficient details, including open source code, to assume the submission is correct, sound and reproducible.

**Documentation:**

An open-source repository and licensing is provided. Further information about each game is provided in the Appendix.

**Limitations:**

The authors discuss the limitation of the lack of Atari support and non-dynamic number of agents based on the particular JAX-based parallelized environments.

**Opportunities For Improvement:**

The library is a strong contribution, however, the paper itself exhibits certain shortcomings. One notable area is the lack of clarity in its motivation. The paper does not effectively communicate the reasons behind the necessity for faster RL environments. It would greatly benefit from concrete examples of problems that can be effectively addressed through the use of faster environments.
There are also discrepancies and inconsistencies in the motivation for the choice of implemented environments. For example, much of the motivation is around board games, but the library also provides five MinAtari games. These are based on pixel based observations and do not seem to fit well with the board game motivation. Furthermore, the paper consistently suggests the focus is on multi-player games, but includes the Atari games and single-player games such as 2048 etc.

A secondary concern with the MinAtari games is that they are already publicly available in  Gymnax – what is the motivation for the re-implementation? What are the differences in implementation?

There are also inconsistencies in the text: Line 73: Gymnax [20] provides more than simply “implement basic RL environments”. For example, they include the suite of MinAtari games. And Jumanji [7] focuses on combinatorial optimisation problems, and also provides some (non-classic) board games, such as snake and 2048.

A notable omission from the paper is a detailed description of the actual PGx API. Instead of providing only a high-level comparison to the Petting Zoo API in Section 3.1, the paper should offer an in-depth explanation of the API. While Figure 2's caption provides some information, such crucial details should be integrated into the main body of the paper. To rectify this, I would suggest adding a subsection that delves into the code snippet, highlighting the differences between this API and other popular APIs. Additionally, it would be beneficial to outline the attributes included in State/observation. For instance, for the Go 19x19 game, the reader is left unclear about the content of the 17 channels within the observation shape (B, 19, 19, 17). This information should be included in the appendix section for each game.
In addition, I would anticipate a thorough high-level discussion, or even more detailed insights, explaining the reasons behind the superior efficiency of on-device processes. This could include a comparison of XLA versus multi-thread parallelization. For instance, it would be valuable to explore the interchangeable use of vectorization and parallelization, which are not equivalent in JAX.
I am curious about how the authors would introduce games where the number of agents dynamically changes. How would you implement this?

Experiments:
In Section 5, what is the rationale behind only evaluating a single agent AlphaZero on the miniature environments (animal shogi, gardener chess, Go 9x9)? I would have expected a more thorough evaluation of a range of agents, on a broad range of different PGx environments - especially given the throughput of the full-sized boardgames is a similar order of magnitude to the toy-environments. Additionally, why is MinAtari only benchmarked using PPO?

Why do the authors not provide baseline models for all environments? For example Go 19x19, Chess and Shogi. This omission makes the benchmark feel unfinished.

Section 6: why is there only a 50% scaling factor when using 8 times the A100. Did you consider an 8 times larger batchsize?

**Relation To Prior Work:**

Yes. Previous works are discussed in 3 clear subsections: Games in AI research; Games as RL envs; Hardware accelerated RL envs; and Algorithms and Architectures that can leverage Pgx.

**Summary And Contributions:**

The authors introduce Pgx, a suite of 20 discrete board game environments, (e.g. Backgammon, Chess, Go and Shogi), written in JAX that can run on GPU/TPUs in parallel and at high throughput. By leveraging auto-vectorization and JIT compilation the environments can be efficiently scaled to thousands of parallel executions. The authors also provide miniature versions of the environments for rapid research cycles. The paper demonstrates experiments 10-100x faster than existing Python RL libraries, such as PettingZoo and OpenSpiel, on a DGX-A100 workstation. They also provide performance benchmarking experiments, and an experiment using AlphaZero on 5 environments.

---

> ### Author Response · Authors · 2023-08-21
> **Author response to reviewer bSTx (1/2)**
>
> Thank you for very detailed and insightful feedbacks.
>
> >  There are also discrepancies and inconsistencies in the motivation for the choice of implemented environments. For example, much of the motivation is around board games, but the library also provides five MinAtari games. These are based on pixel based observations and do not seem to fit well with the board game motivation.
>
> Thank you for pointing out the discrepancies in our motivation for the choice of implemented environments.
> While the core emphasis of Pgx is indeed on board games, we have endeavored to make the library both comprehensive and adaptable. Recognizing the need to clarify the inclusion of MinAtar, we have expanded upon its relevance in Sec. 3.2. We believe this addition will provide a clearer understanding of our intent and the library's versatility.
>
> > Furthermore, the paper consistently suggests the focus is on multi-player games, but includes the Atari games and single-player games such as 2048 etc.
>
> Thank you for highlighting the apparent inconsistency in our focus on multi-player games while also including Atari games and single-player games like 2048.
> The inclusion of the game 2048 serves as a (relatively) straightforward testbed for stochastic yet perfect information games. We have revised the manuscript to justify our choice, referencing previous works that utilized 2048 as a benchmark. Additionally, we have provided further clarification regarding the inclusion of Atari games, emphasizing our aim to showcase the versatility and breadth of our library.
>
> > A secondary concern with the MinAtari games is that they are already publicly available in Gymnax – what is the motivation for the re-implementation? What are the differences in implementation?
>
> The primary reason for our re-implementation is for completeness. Gymnax's MinAtar does not implement Seqauest. However, Seaquest is notably mentioned as an essential testbed for evaluating exploration ability [1]. We have added this clarification to the manuscript.
>
> > There are also inconsistencies in the text: Line 73: Gymnax [20] provides more than simply “implement basic RL environments”. For example, they include the suite of MinAtari games. And Jumanji [7] focuses on combinatorial optimisation problems, and also provides some (non-classic) board games, such as snake and 2048.
>
> Thank you for your feedback. We improved the description for Gymnax and Jumanji in the manuscript.
>
> > A notable omission from the paper is a detailed description of the actual PGx API. Instead of providing only a high-level comparison to the Petting Zoo API in Section 3.1, the paper should offer an in-depth explanation of the API. While Figure 2's caption provides some information, such crucial details should be integrated into the main body of the paper. To rectify this, I would suggest adding a subsection that delves into the code snippet, highlighting the differences between this API and other popular APIs.
>
> Thank you for your valuable feedback. However, there has been criticism that a detailed explanation of the API often resembles library documentation rather than a research paper [2,3]. In response to your comments, we have taken the following steps in the manuscript:
>
> -  In Appendix B, we added the Brax/PettingZoo API code and included a comparison with the Pgx API. We also added references from the main text to the Appendix B.
> -   We incorporated links to our documentation, making it easier for readers to access more detailed specifications and usage instructions.
>
> > Additionally, it would be beneficial to outline the attributes included in State/observation. For instance, for the Go 19x19 game, the reader is left unclear about the content of the 17 channels within the observation shape (B, 19, 19, 17). This information should be included in the appendix section for each game.
>
> We updated our Appendix and provide more detailed description of each environment.
>
> > In addition, I would anticipate a thorough high-level discussion, or even more detailed insights, explaining the reasons behind the superior efficiency of on-device processes. This could include a comparison of XLA versus multi-thread parallelization. For instance, it would be valuable to explore the interchangeable use of vectorization and parallelization, which are not equivalent in JAX.
>
> Thank you for the insightful suggestion regarding a deeper comparison of vectorization and  parallelization.
> Creating environments using C++ threading, like EnvPool, itself requires significant effort. To the best of our knowledge, there is no comprehensive game environments providing C++ threading parallelization (like EnvPool) but available from Python. This is the primary reason we believe a comprehensive comparison between our approach and C++ threading parallelization would be challenging.

---

> > ### Author Response · Authors · 2023-08-21
> > **Author response to reviewer bSTx (2/2)**
> >
> > > I am curious about how the authors would introduce games where the number of agents dynamically changes. How would you implement this?
> >
> > As stated in our limitations section, Pgx API is not a universal one that can adapt to any environment. However, we believe that not every API needs to be all-inclusive.
> > For environments with dynamic agent changes, we believe a different, more suitable API should be developed for that purpose.
> > While our API and its applications have limitations, we believe these limitations themselves make Pgx specialized and user-friendly for the board game domain.
> >
> > > Experiments: In Section 5, what is the rationale behind only evaluating a single agent AlphaZero on the miniature environments (animal shogi, gardener chess, Go 9x9)? I would have expected a more thorough evaluation of a range of agents, on a broad range of different PGx environments - especially given the throughput of the full-sized boardgames is a similar order of magnitude to the toy-environments. Additionally, why is MinAtari only benchmarked using PPO?
> >
> > Thank you for your comments regarding the experiments in Section 5.
> > We acknowledge the importance of a more comprehensive and exhaustive benchmarking. It is indeed a significant area for future work, and we plan to make periodic updates to the repository to address this.
> >
> > > Why do the authors not provide baseline models for all environments? For example Go 19x19, Chess and Shogi. This omission makes the benchmark feel unfinished.
> >
> > Thank you for pointing out the omission of baseline models for certain environments.
> > We recognize the importance of this aspect for future work, and we intend to update the repository accordingly as we progress.
> >
> > > Section 6: why is there only a 50% scaling factor when using 8 times the A100. Did you consider an 8 times larger batchsize?
> >
> > This number was actually 8192 (8 times larger). We improved the description for clarity.
> >
> > ### Reference
> >
> > - [1] Johan Samir Obando Ceron and Pablo Samuel Castro. Revisiting rainbow: Promoting more insightful and inclusive deep reinforcement learning research. In ICML, 2021.
> > - [2] PettingZoo OpenReview discussion: https://openreview.net/forum?id=fLnsj7fpbPI&noteId=vOjT-GimBj6
> > - [3] Brax OpenReview discussion: https://openreview.net/forum?id=VdvDlnnjzIN&noteId=Vs-sOF1edD

---

> > > ### Comment · Reviewer_bSTx · 2023-08-22
> > > **Good updates, still room to improve the experimental side.**
> > >
> > > I thank the authors for engaging in the discussion and providing helpful clarifications and additional information. I have increased my score.
> > >
> > > As per my original comments, I believe that the paper and the repository can be improved with the addition of the experiments/baselines in the future work that you mentioned. However, the updated version of the paper is more clear and readable, and PGx is a valuable contribution to the community.

---

> > > > ### Author Response · Authors · 2023-08-22
> > > >
> > > > Thank you for your constructive feedback and for recognizing the value of Pgx. We appreciate your suggestions and will certainly incorporate the additional developments, experiments, and baselines to Pgx.

---

### Author Response · Authors · 2023-08-21
**General response**

Dear Reviewers,

We sincerely thank you for dedicating your time to review our paper and for offering invaluable feedback. Your constructive comments and insights are greatly appreciated and have been instrumental in guiding the improvements to our manuscript.

In response to the feedback received, we have made the following revisions to the manuscript:

-   In the abstract, we revised the phrase "Python RL libraries" for clarity (in response to reviewer uMKG).
-   In the introduction, we added a link to our headline results (Fig. 3) to emphasize it (in response to reviewer 5n9R).
- In the related work section, we provided more detailed explanations on gymnax and Jumanji (in response to reviwer bSTx).
- In Appendix B, we compared Pgx API with PettingZoo/Brax API using concrete code examples and linked it from Sec. 3.1 (in response to reviewer bSTx).
- In Sec 3.2, we justified our choices for environments (in response to reviewer bSTx).
- In Sec 4.1, we included additional experimental details (in response to reviewer  uMKG).
- In Sec 4.2, we elaborated on the purpose of Fig. 4 in detail (in response to reviewer 5n9R).
- In Sec 4.2, we clarified the self-play batch size hyperparamters (in response to reviewer fFc4).
- In Sec 4.2, we clarified the improvement from a single gpu to eight gpus (in response to reviewer 5n9R and fFc4)
- In Fig. 4, we added horizontal sub-grid lines to emphasize its log-scale (in response to reviewer 5n9R and fFc4).
- In Fig. 5, we plotted the baseline Elo (in response to reviewer  5n9R).
- In Sec. 6, we clarified the self-play batch size hyperparameters (in response to reviewer bSTx).
- In Fig. 6, we adjusted the y-axis limits (in response to reviewer e7SA).
- In Appendix D, we expanded upon the game descriptions (in response to reviewer bSTx).

Changes made in response to your feedback are highlighted in blue.
Also, there are other minor modifications like typo fix.

Once again, thank you for your thoughtful feedback. We believe these revisions have enhanced the quality and clarity of our work.

Best regards,

Authors

---

### Decision · Program_Chairs · 2023-09-22

**Decision:**

Accept (Poster)

**Comment:**

Reviewers have generally provided positive feedback on the paper, acknowledging the significance of PGX as a hardware-accelerated reinforcement learning (RL) environment suite for board games. They appreciate the code's robustness, benchmarking experiments, and the potential it offers to make board game RL more accessible.

Some reviewers raised questions regarding performance, scalability, and the inclusion of real-world tasks. In response to the reviewers' feedback, the authors have made improvements, clarified issues, and provided additional insights, which have enhanced the paper's quality and readability.

To sum up, PGX addresses a significant need in RL research, providing efficient simulators for board games and lowering the barriers to entry for researchers in this domain, which, I believe, can be impactful to the community. In addition, the codebase of PGX is particularly well-developed and can be a great standard and reference for future RL environments.